# Diffusion X-ray image denoising

**Daniel Sanderson**[1,3,5]                                                  DSANDERS@ING.UC3M.ES
**Pablo M.Olmos**[2,3]                                                        PAMARTIN@ING.UC3M.ES
**Carlos Fernández del Cerro**[1,3]                                         CARLOSFE@PA.UC3M.ES
**Manuel Desco**[1,3,4,5]                                                    MANUEL.DESCO@UC3M.ES
**Mónica Abella**[1,3,5]                                                      MABELLA@ING.UC3M.ES

[1] *Departamento de Bioingeniería, Universidad Carlos III de Madrid. Madrid, Spain*

[2] *Departamento de Teoría de la Señal, Universidad Carlos III de Madrid. Madrid, Spain*

[3] *Instituto de Investigación Sanitaria Gregorio Marañón. Madrid, Spain*

[4] *Centro Nacional de Investigaciones Cardiovasculares Carlos III (CNIC), Madrid, Spain.*

[5] *Centro de investigación en red en salud mental (CIBERSAM), Madrid, Spain.*

**Editors:** Accepted for publication at MIDL 2024

## Abstract

X-ray imaging is a cornerstone in medical diagnosis, constituting a significant portion of the radiation dose encountered by patients. Despite the imperative to reduce radiation doses, conventional image processing methods for X-ray denoising often struggle with heuristic parameter calibration and prolonged execution times. Deep Learning solutions have emerged as promising alternatives, but their effectiveness varies, and challenges persist in preserving image quality. This paper presents an exploration of diffusion models for planar X-ray image denoising, a novel approach that to our knowledge has not been yet investigated in this domain. Evaluation on clinical data shows that our approach enables real time denoising of Poisson noise while preserving image resolution and structural similarity. This suggests that diffusion models are promising for planar X-ray image denoising, offering a potential improvement in the optimization of diagnostic utility amid dose reduction efforts.

**Keywords:** X-ray, radiography, dose, denoising, diffusion model.

## 1. Introduction and related work

X-ray imaging accounts for 93.7% of the mean radiation dose applied to patients in medical diagnosis (Nuclear, 2010). A single X-ray chest image implies an approximate effective radiation dose of 0.1 mSv, which is equivalent to 10 days of exposure to ambient radiation. For fluoroscopic interventions or clinical studies requiring several planar X-ray images, doses can significantly build up, posing a risk for the patient, especially for paediatric patients (Luo et al., 2020). Additionally, large radiation doses can lead to premature hardware failure of the X-ray equipment, due to vaporization of the tube´s anode and breakdown of the tube´s filament. Therefore, it is important to reduce the dose of X-ray acquisitions. However, a reduction in dose implies an increase in image noise, which arises due to the quantic nature of X-rays and the presence of thermal fluctuations in the detector (Ding et al., 2018; Yi and Babyn, 2018). This hampers the contrast resolution of the image, limiting the diagnostic utility of the radiography and degrading the performance of downstream processing or feature extraction algorithms (Juneja et al., 2023).

Several conventional image processing methods have been proposed to perform denoising of X-ray images. Some of the methods that have shown good results are bilateral filters (Juneja et al., 2023), total variation (TV) methods (Sagheer and George, 2020), or 3D Block Matching (Dabov et al., 2007). However, these methods require heuristic calibration of parameters hindering their generalization and can have long execution times limiting their incorporation into clinical practice (Lin et al., 2023). To solve these limitations, several Deep Learning (DL) solutions have been proposed for planar X-ray and fluoroscopy image denoising. To date most DL works simulate Gaussian noise, despite being Poisson noise the most relevant type of noise in X-ray images (Ding et al., 2018; Yi and Babyn, 2018).

Depending on the type of data used during training, most DL solutions applied to X-ray denoising can be classified into two main categories. The first aim to predict the clean image from the noisy image using simple networks such as DnCNN or Denoising Autoencoders (Juneja et al., 2023; Gondara, 2016) or heuristically designed architectures composed of feature extracting and refinement blocks (Nayak et al., 2023) or of dual denoising networks(Sahu et al., 2023). The second category of methods either train the network with pairs of noisy images of different noise content (Noise2Noise methods) or aim to predict specific pixels selected either randomly (Krull et al., 2019) or by intensity thresholding (Batson and Royer, 2019) (Noise2Self methods). The majority of these solutions minimize MSE estimates of the target, such as the Charbonnier or Frobenius norm, while only a few explore alternative loss functions (Matviychuk et al., 2016). As MSE estimates only compare pixel wise differences, it is common to obtain results of reduced perceptual quality, generally leading to a loss of spatial resolution (Chung et al., 2022a) or to incomplete noise removal.

To better preserve image resolution and texture, generative models such as Generative Adversarial Networks (GANs) have been recently been applied to medical image denoising. However, GANs suffer from convergence issues, mode collapse and vanishing gradients, greatly hindering their training (Skandarani et al., 2023). Recently, diffusion models have outperformed GANs (Dhariwal and Nichol, 2021), and have further improved image quality with notably simple models. These models apply a Noise2Noise training strategy, simulating noise in a self-supervised fashion at different noise levels and predicting the residual. Diffusion models are recently being applied to CT denoising (Xia et al., 2022; Liu et al., 2023), but to our knowledge they have not yet been applied to denoising of planar X-ray and/or fluoroscopy images.

In this work, we propose a denoising method based on diffusion models for planar X-ray imaging. The method is trained with a small database to mimic the conditions of clinical scenarios where images are difficult to obtain and is evaluated on images contaminated with Poisson noise.

## 2. Materials and Methods

The proposed method is based on the original implementation of Denoising Diffusion Probabilistic Models (DDPMs) (Ho et al., 2020), which are designed for generative modelling by using a DL network to sequentially remove noise in a residual fashion.

Figure 1 shows the workflow of the proposed method, DDPM-X, that consists of two stages: I) a diffusion model is trained with real clinical data for image generation by pro-

gressively eliminating Gaussian noise starting from pure Gaussian noise and II), the method identifies the step of the generative pipeline from which to start denoising real images. This is achieved by identifying the specific denoising step within the generative pipeline that corresponds to the equivalent noise level of the noisy image. For this stage we used real images with simulated noise.

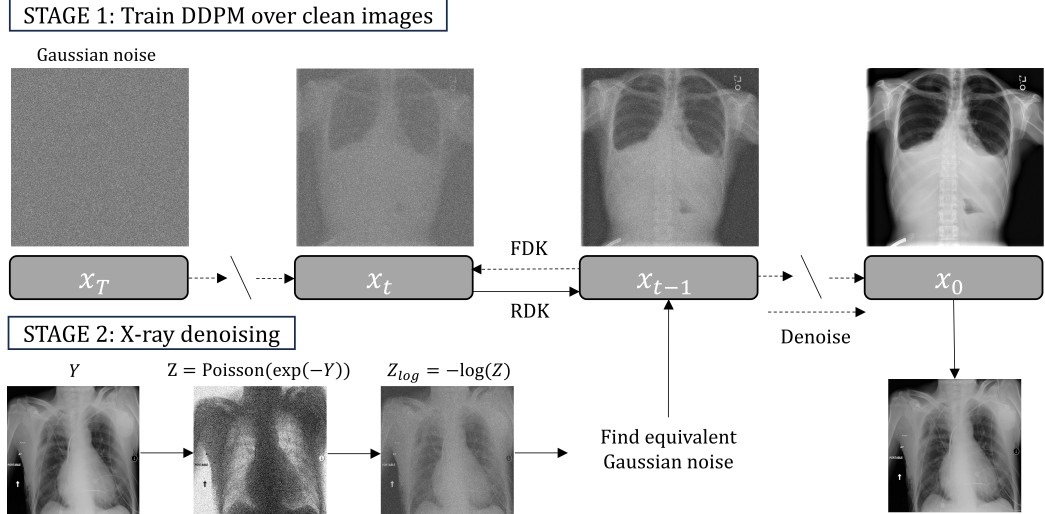

Figure 1: Workflow of the proposed method.

Evaluation was done on images contaminated with Gaussian noise, for which the DDPM-X was trained, and Poisson noise, which better models the noise found in X-ray images.

### 2.1. Generative diffusion model

As shown in Figure 1, the diffusion model used in DDPM-X consists of three elements: a Forward Diffusion Kernel (FDK), a DL network, and a Reverse Diffusion Kernel (RDK). The FDK iteratively applies a diffusion process to corrupt a clean image $x_0 \in \Re^{H \times W}$ with Additive White Gaussian Noise (AWGN) for a set of timesteps $t \in [0, T]$, as shown in Equation 1.

$$q(x_t|x_{t-1}) = \mathcal{N}(x_t; \sqrt{1 - \beta_t}x_{t-1}, \beta_t \mathbf{I}) \tag{1}$$

where $x_t$ is the noise corrupted image for timestep $t$ and $\beta_t$ is a hyperparameter. The FDK is applied in a single step for any random $t$ as follows:

$$q(x_{0:t}|x_0) = q(x_1, x_2, ..., x_t|x_0) \overset{Markov}{=} \prod_{t=1}^{T} q(x_t|x_{t-1}) = \mathcal{N}(x_t; \sqrt{\overline{\alpha_t}}x_0, (1 - \overline{\alpha_t})\mathbf{I}) \tag{2}$$

where $\alpha_t = 1 - \beta_t$ and $\overline{\alpha_t} = \prod_{s=0}^{t} \alpha_s$. $\beta_t$ may be updated following any differentiable function that ensures that $\sqrt{\overline{\alpha_T}} \approx 0$. We use the linear function presented in (Ho et al., 2020): $\beta_t = \frac{\beta_e - \beta_s}{T}t + \beta_s$, with $\beta_s = 10^{-6}$ and $\beta_e = 0.02$. The value of $\beta_s$ was selected to ensure that evaluation noise levels corresponded to realistic X-ray doses.

The RDK generates an image by iteratively reversing the forward diffusion. Given that the reversal of a Gaussian diffusion process is also Gaussian, the p.d.f of the data can be recovered by marginalization of the individual Markov steps:

$$p_\theta(x_0) \overset{marginal}{=} \int p_\theta(x_{0:T})\partial x_{1:T} \overset{markov}{=} \int p(x_T) \prod_{t=0}^{T-1} p_\theta(x_{t-1}|x_t)\partial x_{1:T} \tag{3}$$

where $\theta \in \Re^\Theta$ are the parameters of the DL network, $p(x_T) \sim \mathcal{N}(0,\mathbf{I})$, and $p_\theta(x_{t-1}|x_t) = \mathcal{N}(x_{t-1}; \mu_\theta(x_t,t), \Sigma_\theta(x_t))$ is a step from the RDK, being $\mu_\theta \approx x_t - \eta_\theta$ and $\eta_\theta$ the Gaussian noise prediced by the network. For simplicity, we used a fixed small variance $\Sigma_\theta = \overline{\beta_t}$. At each step of the RDK, we clipped $\mu_0$ to the [-1,1] intensity range (Saharia et al., 2022).

## 2.2. Denoising strategy

To perform denoising we follow a similar approach to the Come Closer Diffuse Faster algorithm (CCDF) (Chung et al., 2022b). The RDK is applied from $t = t'$ to $t = 0$ (Fig. 1), where $t'$ is the denoising timestep obtained from an estimate of the noise level of the image. The approach followed to compute the denoising timestep $t'$ varies depending on the probabilistic model used to simulate noise. In this work we consider two noise models: Gaussian noise, for which the DL network has been specifically trained, and Poisson noise, which is the type of noise inherent to X-rays due to their quantic nature. Gaussian noise is simulated with equation 2 for a specific timestep $t$, and therefore $t' = t$. Poisson noise is simulated using equation 4, which includes a small Gaussian noise $\eta \sim \mathcal{N}(0,\mathbf{I})$ scaled by $\sigma^2 = 10$ to emulate electronic noise, as in (Gao et al., 2023).

$$Z_{log} = -log\left(\frac{Poisson(Y) + \sigma^2\eta}{I}\right) \tag{4}$$

where $Y = Ie^{-Y_{log}}$, $Y_{log}$ is the noiseless image, and $\int I_0(\epsilon)\partial\epsilon = I$ represents the flood image. Due to the signal dependency of Poisson noise, the denoising timestep $t'$ is estimated from the maximum noise variance found in the image, as follows:

$$\hat{Y_{log}} = \mathbf{I} * \frac{1}{n} \sum_{i=1}^{N} P_{99}(\max(Y_{log}^i)), \quad \mathbf{I} \in \mathbb{R}^{H \times W} \tag{5a}$$

where $N$ correspond to the size of the training dataset. The percentile is applied to avoid the contribution of high intensity artificial details present in the images such as medical annotations. Then, $\hat{Z}_{log}$ is computed from $\hat{Y} = Ie^{-\hat{Y}_{log}}$ by using Equation 4, and the denoising timestep $t'$ is estimated as follows:

$$t' := 1 - \overline{\alpha'_t} \approx \widehat{\text{Var}}(\hat{Z}_{log}) \tag{6}$$

Calculating the timestep $t'$ using Equation 6 ensures that the model removes the noise of maximum variance. It must be noticed that obtaining $Z_{log}$ requires knowing the dose $I$. For real noisy images, $I$ can be estimated from the X-ray acquisition parameters, or by using noise estimation methods (Turajlić and Karahodzic, 2017). To ensure an equivalent noise level between Gaussian and Poisson noise, Gaussian noise was simulated by taking the timestep $t$ from Equation 2 as the denoising timestep $t'$ estimated for Poisson noise.

## 2.3. Network

We used a U-Net composed of five pairs of downsampling and upsampling blocks with SiLU activation functions, each built of 2 Resnet layers, and an attention block of 8 heads. The number of output channels per downsampling and attention block were duplicated from 128 to 512 every two blocks. The network was conditioned on the timestep which was given to each block as a sinusoidal embedding preprocessed by an MLP block of two layers.

The model was trained for 100 epochs with mixed precision, using a learning rate of $10^{-4}$, an AdamW optimizer, and a cosine schedule to achieve super convergence (Smith and Topin, 2019). The MSE loss function $L(\theta) = \mathbb{E}[\|\eta - \eta_\theta(x_t, t)\|^2]$ was used to predict the Gaussian noise $\eta$ of the image at a timestep $t$ randomly drawn from a uniform distribution. A random horizontal flip was applied to the images to perform data augmentation. Training was performed on a RTX 3090 GPU of 24 GB, and took 500 s per epoch, while inference took 0.25s per timestep. All code was implemented on Pytorch based on Fastai (Howard and Gugger, 2020) and Diffusers from Hugging Face (von Platen et al., 2022).

## 2.4. Evaluation

The proposed method was evaluated for Poisson noise and Gaussian equivalent noise. Noisy images were obtained for high dose with $I = 5 \times 10^4$ and $\sqrt{1 - \overline{\alpha_3}} = 9.6 \times 10^{-3}$ for Poisson and Gaussian noise, respectively, and for low dose with $I = 9 \times 10^3$ and $\sqrt{1 - \overline{\alpha_9}} = 9.6 \times 10^{-2}$, for Poisson and Gaussian noise, respectively. The high dose corresponded to an estimated denoising step of $t' = 3$ and the low dose to $t' = 9$. We randomly selected 1225 images from the NIH Chest X ray database (Wang et al., 2017), splitted into a training set of 1125 images and a validation set of 100 images. Images were resized from 1024x1024 to 512x512 and normalized to the [-1,1] intensity range, as in (Matviychuk et al., 2016). These images were taken as the noiseless image $Y_{log}$ from Equation 4.

The evaluation of Poisson contaminated images was compared with four well-known algorithms: Block Matching and 3D filtering (BM3D) (Dabov et al., 2007); Neighbor2Neighbor (Nei2Nei) (Huang et al., 2021a); Dual GAN (DU-GAN) (Huang et al., 2021b), and the same UNet architecture used by DDPM-X trained in a supervised fashion with the MSE loss. As BM3D is designed for Gaussian noise, to be fair we preprocessed the images with the Anscombe transform to convert Poisson noise into Gaussian of variance 1 and normalized them to the [0,1] intensity range as in (Bodduna and Weickert, 2019).

To evaluate the performance of the models, we applied three metrics commonly used in denoising: Peak Signal to Noise Ratio (PSNR), to evaluate pixelwise differences, Learned Perceptual Image Patch Similarity (LPIPS), to evaluate visual quality, and Structural Similarity Index Measure (SSIM), to evaluate both distortion and visual quality (Blau and Michaeli, 2018). The absolute difference (AD) was obtained as the difference between the metrics computed for the target and the denoised image, and the relative difference (RD) as the ratio of AD and the metric computed for the target and noisy image. Visual evaluation was done after a simple post-processing pipeline, consisting of a Contrast Limited Adaptive Histogram Equalization (CLAHE) with size tile of 70 píxels and clip limit of 0.0001, and a Laplacian Pyramid of 3 levels. We additionally performed a preliminary evaluation of our method with real noisy data acquired at a low dose. This evaluation is found in Appendix A.

## 3. Results

Table 1 shows that our method is powerful enough to achieve significant denoising for a wide range of noise levels.

Table 1: Metrics evaluated on the validation set at different dose levels.

| Dose | AD | | | RD | | |
|---|---|---|---|---|---|---|
| I ($\times 10^3$) | LPIPS | SSIM | PSNR | LPIPS | SSIM | PSNR |
| 9 | 0.02 | 98.05 | 36.89 | **92.41** | **12.68** | **22.63** |
| 14 | 0.02 | 98.25 | 38.36 | 91.93 | 7.74 | 19.57 |
| 33 | 0.01 | 98.90 | 40.97 | 90.59 | 3.06 | 14.37 |
| 50 | 0.01 | 99.06 | 41.93 | 87.07 | 1.86 | 11.50 |
| 100 | **0.00** | **99.47** | **44.37** | 78.75 | 0.87 | 9.23 |

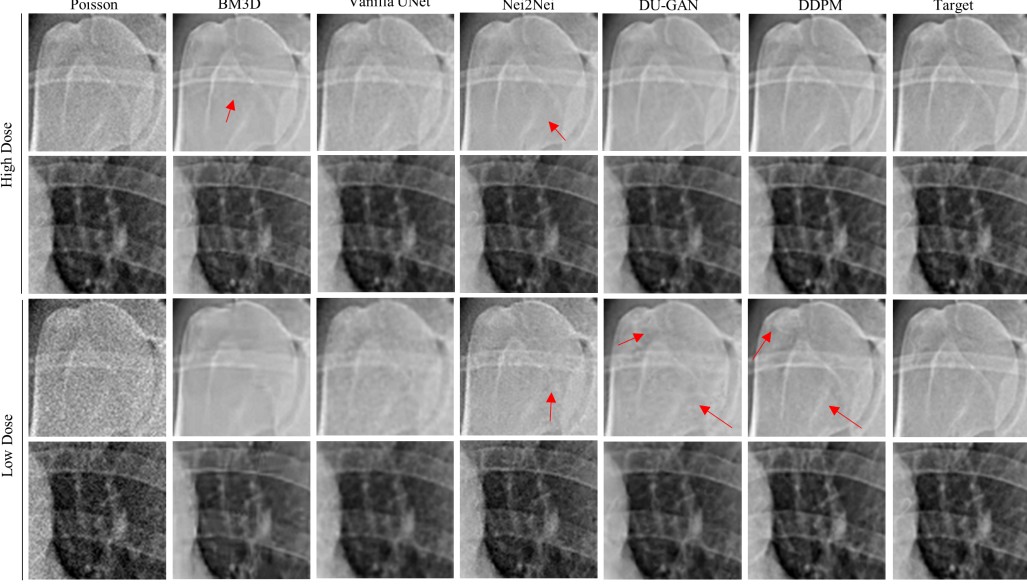

Figure 2: Zoom of shoulder and lung regions indicated by the yellow rectangles of Figure 4. Red arrows point to hallucinations.

Table 2 shows that DDPM-X achieved the best quantitative performance of all methods for most. We can see that for most metrics, and specially for low doses, the best results were achived by DDPM-X for Gaussian noise, for which the network was trained. However, the performance for Poisson noise was similar, with percentually small differences in RD (below 3% in the worst of the cases) and from a qualitative point of view, these differences do not significantly hinder the visualization of the denoised image (Figure 2). The denosing timesteps for which the best quantitative results were obtained shows an error in the estimations of $t' = \pm 1$ for our method ($t'$ in Table 2).

Visual results in Figure 2 show that DDPM-X can effectively handle the different Poisson noise levels found in the images due to the signal dependency of Poisson noise, while preserving spatial resolution. However, for low doses it was unable to restore details of low contrast resolution that had been masked by noise, and in some cases it introduced a slight spatial distortion and/or small details as shown by the red arrows. BM3D introduces smoothing and artificial textures, being more noticeable for low doses, while Nei2Nei loses low contrast details for high doses and fails to achieve complete denoising for low doses. The vanilla UNet blurs the images for both high and low doses, while the DU-GAN preserves spatial resolution for high doses but introduces smoothing for low doses.

Figure 3 shows that for denoising timesteps above the optimum, the value of the metrics for DDPM-X on Poisson noise is not significantly affected. However, visual evaluation shows smoothing and hallucinations (Figure 4). For smaller timesteps, images preserve spatial content despite suffering from incomplete noise removal.

Table 2: Quantitative results of the models for $t'$ denoising steps. DDPM-Xg and DDPM-Xp correspond to DDPM-X evaluated on Gaussian and Poisson noise respectively. **Best results**, second best results

| Dose | Model | t' | LPIPS ↓ | | SSIM ↑ | | PSNR ↑ | |
|------|-------|-----|------|------|------|------|------|------|
| | | | AD | RD | AD | RD | AD | RD |
| Low | BM3D | - | 0.05 | 78.69 | 97.81 | 11.81 | 34.58 | 22.38 |
| | Vanilla UNet | - | 0.07 | 72.42 | 97.81 | 12.59 | **37.56** | 20.25 |
| | Nei2Nei | - | 0.03 | 85.01 | 96.85 | 11.14 | 36.45 | 16.94 |
| | DU-GAN | - | 0.04 | 81.79 | 97.61 | 12.27 | 37.04 | 19.48 |
| | DDPM-Xg | 8 | **0.02** | **93.81** | 97.33 | **14.67** | 37.27 | **26.49** |
| | DDPX-Xp | 8 | 0.02 | 92.40 | **98.05** | 12.68 | 36.89 | 22.63 |
| High | BM3D | - | 0.02 | 65.88 | 98.88 | **1.99** | 39.78 | **12.11** |
| | Vanilla UNet | - | 0.02 | 64.66 | 98.68 | 1.81 | 41.39 | 7.21 |
| | Nei2Nei | - | 0.02 | 53.21 | 98.80 | 1.71 | 40.45 | 5.90 |
| | DU-GAN | - | 0.01 | 84.06 | 99.11 | 1.88 | 42.23 | 9.24 |
| | DDPM-Xg | 3 | **0.01** | **89.87** | **99.15** | 1.63 | **42.54** | 11.61 |
| | DDPM-Xp | 4 | 0.01 | 87.06 | 99.07 | 1.86 | 41.93 | 11.50 |

## 4. Discussion

In this work we have proposed DDPM-X, a method for planar X-ray image denoising based on a diffusion model. Although the network was trained on Gaussian noise, results suggest that the diffusion model can be also applied to Poisson denoising without any modification or fine-tuning of the network.

Given the noise level conditioning of the network, the user can regulate the amount of denoising ad hoc. An initial estimate of the noise level, which is used to define the denoising timestep, may be obtained from the SVD decomposition of the image (Turajlić and Karahodzic, 2017), or from the X-ray dose associated to the acquisition parameters.

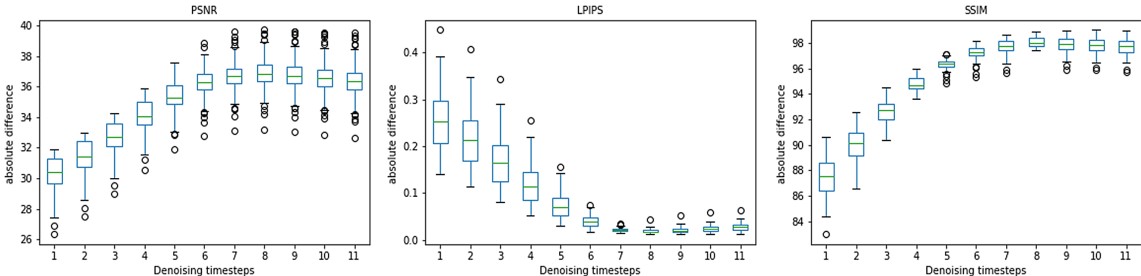

Figure 3: Mean values of the metrics for different denoising steps for the low dose case on the validation set.

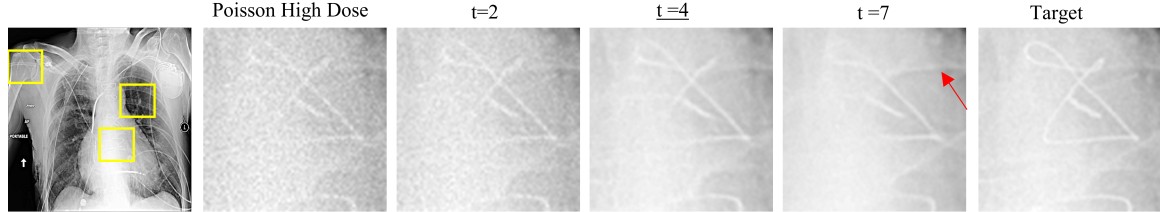

Figure 4: Denoising results of DDPM-X for Poisson noise for different timesteps, shown for the spine region indicated by the yellow rectangle. Arrow points to hallucination.

However, the selection of the appropriate denoising timestep can be critical as overly large values can introduce hallucinations. Given that the preservation of spatial content is of uttermost importance in the medical field, it is therefore preferable to cautiously use smaller values. In the future, we will explore the inclusion of data consistency models to constrain the generative power of these models and reduce the risk of content distortion.

Evaluation showed that the metrics failed to detect the appearance of hallucinations. In the future we will explore alternative metrics such as the Edge Preservation Index (EPI) (Sagheer and George, 2020), and we will evaluate them patch-wise to account for small local spatial distortions. On the other hand, the low differences in metric values between our method and the baselines did not seemingly correlate with the noticeable visual differences observed. This discrepancy is likely caused by the difference in contrast between the denoised image used to evaluate the metrics and the contrast-enhanced images used for visualization.

The proposed method can be efficiently trained with a small database of down to approximately 1100 images, enabling its application to real clinical scenarios which often lack large databases. Inference can be done in less than 3 seconds allowing its real-time application. The method could be further sped up by estimating the variance $\Sigma_\theta$ of the reverse diffusion path rather than taking a fixed value (Song et al., 2020).

## Acknowledgments

This work was supported by PDC2021-121656-I00 (MULTIRAD) and PID2021-123182OB-I00, funded by MCIN/AEI/ 10.13039/501100011033 and by the 'NextGenerationEU'/PRTR and European Union 'FEDER'. Also funded by Instituto de Salud Carlos III through the projects PT20/00044, co-funded by the European Regional Development Fund "A way to make Europe" and PMPTA22/00121 and PMPTA22/00118, co-funded by the European Union 'NextGenerationEU'/PRTR, and by Comunidad de Madrid under grants IND2022/TIC-23550 and ELLIS Unit Madrid.The CNIC is supported by Instituto de Salud Carlos III, Ministerio de Ciencia e Innovación, and the Pro CNIC Foundation.

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

## Appendix A. Evaluation on real noise

DDPM-X was evaluated on a real noisy acquisition of an anthropomorphic phantom. Target images were acquired at 100kV and 4 mAs, while noisy images were acquired at the same voltage and 0.8 mAs. As can be seen in Figure 5, results are slightly blurry, especially in the lung region. This may be because the phantom image has perfect borders of high resolution and deviates from the data distribution on which the model was trained. Table 3 shows that quantitative results are almost identical than for simulated data excepting PSNR, which is surprisingly low likely due to a non perfect alignment of the phantom for the low and high dose acquisitions. Despite this, results show the promise of our model on real data.

Table 3: Quantitative results of DDPM-X for $t'$ denoising steps.

| Denoise timestep $t'$ | LPIPS | MSSIM | SSIM | PSNR |
|---|---|---|---|---|
| 4.00 | 0.02 | 99.77 | **98.15** | 27.87 |
| 3.00 | **0.02** | **99.78** | 98.15 | **27.88** |
| 2.00 | 0.02 | 99.77 | 98.00 | 27.87 |

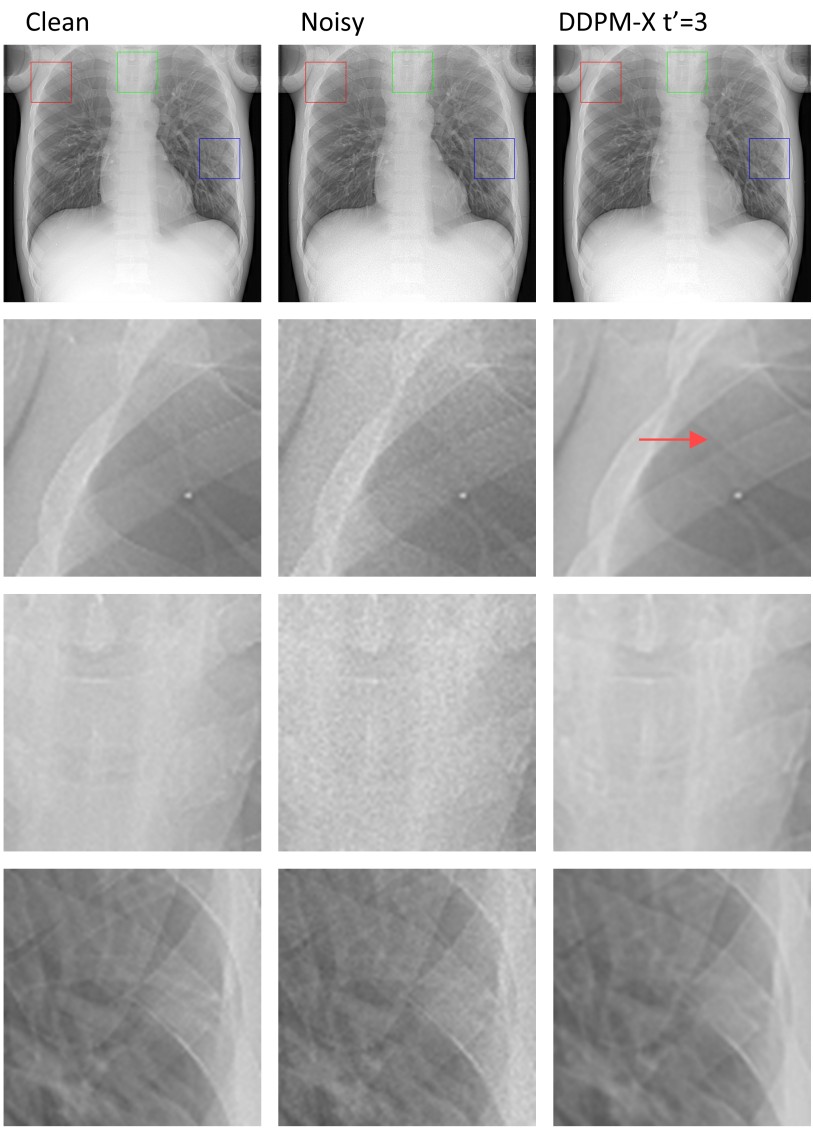

Figure 5: Denoising results for an antropomorphic phantom acquired at 100 kV/ 4 mAs and 100kV/0.8 mAs. Red arrow points to missing details.

