# OpenReview forum: "Diffusion X-ray image denoising"
_MIDL.io/2024/Conference — MIDL 2024 Poster_

### Official Review · Reviewer_sYEE · 2024-02-21

**Confidence:** 4
**Preliminary Rating:** 2
**Recommendation:** Poster
**Final Rating:** 4

**Summary:**

In this paper, a diffusion-model-based method is proposed to denoise X-ray. The author stated that this is the first study that applies the denoise diffusion model to X-ray denoise. The validation experiment on a simulated dataset demonstrates the proposed method outperforms the two existing methods.

**Strengths:**

1. The setting that applying the diffusion model on X-ray denoise is novel.
2. The experiment demonstrates the effectiveness of the proposed method.
3. The presentation is well-organized and easy to follow.

**Weaknesses:**

1. It seems there already is some work that applies the diffusion model on X-ray denoise. such as
"Cascaded Latent Diffusion Models for High-Resolution Chest X-ray Synthesis" https://link.springer.com/chapter/10.1007/978-3-031-33380-4_14

2. The validation was carried out on simulated data. Although the simulation process has strong theoretical support, the simulated data cannot represent the distribution of real X-ray data.

**Detailed Comments:**

The comparison methods should include more state-of-the-art methods. Currently, only two methods published in 2007 and 2021 were included. Since the author stated the advantage of the diffusion model compared with GAN, GAN-based denoise/reconstruction methods should be included.

The performance of the two existing methods, BM3D and Nei2Nei, is already high, based on the SSIM and PSNR. Does this mean the task is not so challenging?

What is the reason that the denoise performance decreases in the last denoising timesteps (like after the 8th step) in Figure 4.

**Justification Of Final Rating:**

The authors clarified the novelty, used an anthropomorphic phantom as a surrogate, and added the experiments to compare with the state-of-the-art GAN methods, which makes me increase the rating. Thanks the authors for their efforts.

**Justification Of The Preliminary Rating:**

This paper's major contribution is applying the diffusion model to X-ray denoise. Considering there is already some work being published, the method needs some other contributions to meet the accept condition.

**Questions To Address In The Rebuttal:**

The author may need to soften the statement that this is the first work that applies the diffusion model to X-ray denoise.

The author also needs to discuss the contribution of this paper compared with existing methods, including X-ray denoise and other similar topics.

**Special Issue:**

No

---

> ### Author Response · Authors · 2024-03-15
>
> > Q: Already existing realted work, such as "Cascaded Latent Diffusion Models for High-Resolution Chest X-ray Synthesis"
>
> A: We apologize for the confusion. We have rewriten Section 3 to clarify the novelty of our method. The work that the reviewer points to only focuses on high-resolution X-ray image generation and does not perform denoising of real images. Denoising is the working mechanism of diffusion models but should not be confused with actual image denoising, especially when dealing with Poisson noise. In the future, we may consider introducing into Stage 1 of our work some of the strategies presented in that paper. However, to our knowledge, there are still no publications regarding the pure denoising of planar X-ray images utilizing diffusion models.
>
> > Q: The validation was carried out on simulated data, which cannot represent the distribution of real X-ray data.
>
> A: We would like to note that all denoising experiments in Stage II and hence the results in Tables 1 and 2 have been carried out using real X-ray data with simulated noise. Validation of Stage 1 was only carried out to demonstrate that the generative model had been correctly trained, but all data used in Stage 2 was obtained from the NIH Chest Xray database. We have removed Figure 1 to avoid confusion. Nevertheless, we agree with the reviewer that the algorithm should be tested on real noise. However, it is not straightforward to gain access to human X-rays given the ethical concerns, and to our knowledge, there is no public database hosting non-simulated noisy images, neither for CT nor planar X-rays. Therefore, we have used an anthropomorphic phantom as a surrogate. Target images were acquired at 100kV and 4 mAs, while noisy images were acquired at the same voltage and 0.8 mAs. [As can be seen](https://imgbox.com/LcJkaHZq) , results are slightly blurry, especially in the lung region. This may be because the phantom image has perfect borders of high resolution and deviates from the data distribution on which the model was trained. Quantitative results are almost identical than for simulated data excepting PSNR, which is surprisingly low likely due to a non perfect alignment of the phantom for the low and high dose acquisitions. Despite this, we believe that the results are highly positive.
>
> | Denoise timestep | LPIPS    | MSSIM     | SSIM      | PSNR      |
> |------------------|----------|-----------|-----------|-----------|
> | 4.00             | 0.02     | 99.77     | **98.15** | 27.87     |
> | 3.00             | **0.02** | **99.78** | 98.15     | **27.88** |
> | 2.00             | 0.02     | 99.77     | 98.00     | 27.87     |
>
> **Results on real data have been added to the Appendix.**
>
> > Q: GAN baseline.
>
> A: We have included DU-GAN [1], which was originally developed for CT Denoising. We have kept all training hyperparameters unchanged. Given the limited time of the rebuttal period, we only show the validation for low dose, but we will include the high dose case by the end of the discussion period.
>
> From a [qualitative point of view](https://imgbox.com/CGjvOYj8), DU-GAN achieves better denoising than Nei2Nei, and although it does not smooth the image as much as BM3D, it still introduces blurring, specially in low contrast borders. This is likely the cause for the high value of LPIPS shown in the table below. Importantly, DU-GAN introduces halluciantions, as indicated by the red arrows.
>
> | Model   | t' | LPIPS    | SSIM      | PSNR      |
> |---------|----|----------|-----------|-----------|
> | BM3D    | -  | 0.05     | 97.81     | 34.58     |
> | Nei2Nei | -  | 0.03     | 96.85     | 36.45     |
> | DU-GAN  | -  | 0.04     | 97.61     | 37.04     |
> | DDPM-Gaussian | 8  | **0.02** | 97.33     | **37.27** |
> | DDPM-Poisson | 8  | 0.02     | **98.05** | 36.89     |
>
> **DU-GAN baseline will be included once the validation for the high dose case is completed**
>
> > Q: High performance of baselines.
>
> A: Metric values are high because they are being evaluated on the whole image. Given that images are big and have a non-negligible amount of scatter and, therefore a low contrast resolution, it is not surprising that the metrics fail to reflect how truly inferior the performance of the baselines is. As shown in Figure 2 of the paper, differences in performance become especially noticeable from a visual point of view when zooming into the contrast-enhanced image, as any radiologist is expected to do. Errors in low contrast resolution details may go unnoticed by the metrics, but are of great importance for the radiologists. Differences in metric values would be larger if evaluated on the enhanced images, what we decided not to do to avoid any interference of the post-processing algorithm. Nevertheless, we could change this if the reviewer deems it appropriate.
>
> **We have expanded the paper with this discussion.**
>
> ---
> [1] Zhizhong Huang et al. Du-gan: Generative adversarial networks with dual-domain u-net-based discriminators for low-dose ct denoising. 2022

---

> ### Author Response · Authors · 2024-03-15
> **Official comment by Authors part 2**
>
> > Q: Decrease of performance in the last denoising timesteps.
>
> A: The reason is that the model introduces over-smoothing and removes textures. Furthermore, the greater the number of timesteps the model is allowed to act, the more hallucinations it introduces. This was explained in the Results section as follows: *Figure 3 shows that for denoising timesteps above the optimum, the value of the metrics for DDPM-X on Poisson noise is not significantly affected. However, visual evaluation shows smoothing and hallucinations (Figure 4).*
> Further suggestions on how to improve this discussion are welcomed.
>
> > Q: The author also needs to discuss the contribution of this paper compared with existing methods, including X-ray denoise and other similar topics.
>
> A: The current study does not focus on X-ray image generation but denoising. Presently, we're unaware of any research that employs diffusion models exclusively for denoising planar X-ray images. Other works focus solely on generating high-resolution X-ray images, lacking actual image denoising capabilities, particularly against Poisson noise. Therefore, we believe that this work may be of great interest for the medical imaging community.

---

> > ### Author Response · Authors · 2024-03-27
> > **Update**
> >
> > **DU-GAN and vanilla UNet have been included in the manuscript**. The UNet selected was the same as the one used by the proposed method, trained in a supervised fashion with the MSE loss. We have included the quantitative results in Table 2 and the qualitative results in Figure 2.

---

### Official Review · Reviewer_f2Q9 · 2024-02-27

**Confidence:** 4
**Preliminary Rating:** 3
**Final Rating:** 4

**Summary:**

In the broadly practical usage of X-ray, and the sufferings that it faces in maintaining the high quality of image nowadays, the authors aspired to use a diffusion model to denoise the X-ray images. The results show real time denoising of Poisson noise while preserving image resolution and structure similarity

**Strengths:**

- Gaussian and Poisson distributions of noises are studied in this work, performing noises that are frequently seen in the real world.
- The proposed method is proved to be applicable in clinical scenarios where clear images are hard to retrieve.
- Good performance gains.

**Weaknesses:**

- The introduction part of the article is a combination of Introduction and Related works. The authors should really split it into two parts for better clarification.
- The authors imitate two different sorts of noises: Poisson and Gaussian. While the previous works are mostly focused on Gaussian Noises, the authors didn’t provide the reasons for choosing Poisson Noise to denoise.
- The proposed method was only tested on one dataset: NIH Chest X ray database, with a random selection of 1225 images in total. There are two concerns regarding this: 1. The authors should try on different datasets. 2. Why not use the full dataset instead of sampling?
- Are the training timestep and the testing timestep identical? e.g. 3 and 4 in low dose, and 8 in high dose?
Is there any possibility to combine the two models regarding Gaussian and Poisson noises together to perform a better result? So that in real use, we don’t have to choose from these two.

**Detailed Comments:**

- The introduction part of the article is a combination of Introduction and Related works. The authors should really split it into two parts for better clarification.
- The authors imitate two different sorts of noises: Poisson and Gaussian. While the previous works are mostly focused on Gaussian Noises, the authors didn’t provide the reasons for choosing Poisson Noise to denoise.
- The proposed method was only tested on one dataset: NIH Chest X ray database, with a random selection of 1225 images in total. There are two concerns regarding this: 1. The authors should try on different datasets. 2. Why not use the full dataset instead of sampling?
- Are the training timestep and the testing timestep identical? e.g. 3 and 4 in low dose, and 8 in high dose?
Is there any possibility to combine the two models regarding Gaussian and Poisson noises together to perform a better result? So that in real use, we don’t have to choose from these two.

**Justification Of Final Rating:**

Thanks the authors for the classification! The authors have addressed all my concerns regarding its novelty and clarity. In addition, the authors seem to have missed some relevant literature. Specifically, they don't discuss learning-based methods for denosing tasks, missing out on several relevant citations, e.g. “Structurally-Sensitive Multi-Scale Deep Neural Network for Low-Dose CT Denoising”, and “CT super-resolution GAN constrained by the identical, residual, and cycle learning ensemble (GAN-CIRCLE)”. These methods are relevant to the method proposed in this paper. These relevant papers should be discussed in the papers.

**Justification Of The Preliminary Rating:**

- The introduction part of the article is a combination of Introduction and Related works. The authors should really split it into two parts for better clarification.
- The authors imitate two different sorts of noises: Poisson and Gaussian. While the previous works are mostly focused on Gaussian Noises, the authors didn’t provide the reasons for choosing Poisson Noise to denoise.
- The proposed method was only tested on one dataset: NIH Chest X ray database, with a random selection of 1225 images in total. There are two concerns regarding this: 1. The authors should try on different datasets. 2. Why not use the full dataset instead of sampling?
- Are the training timestep and the testing timestep identical? e.g. 3 and 4 in low dose, and 8 in high dose?
Is there any possibility to combine the two models regarding Gaussian and Poisson noises together to perform a better result? So that in real use, we don’t have to choose from these two.

**Questions To Address In The Rebuttal:**

- The introduction part of the article is a combination of Introduction and Related works. The authors should really split it into two parts for better clarification.
- The authors imitate two different sorts of noises: Poisson and Gaussian. While the previous works are mostly focused on Gaussian Noises, the authors didn’t provide the reasons for choosing Poisson Noise to denoise.
- The proposed method was only tested on one dataset: NIH Chest X ray database, with a random selection of 1225 images in total. There are two concerns regarding this: 1. The authors should try on different datasets. 2. Why not use the full dataset instead of sampling?
- Are the training timestep and the testing timestep identical? e.g. 3 and 4 in low dose, and 8 in high dose?
Is there any possibility to combine the two models regarding Gaussian and Poisson noises together to perform a better result? So that in real use, we don’t have to choose from these two.

---

> ### Author Response · Authors · 2024-03-15
>
> > Q: Split Introduction into Introduction and Related works.
>
> A: We thank the reviewer for the suggestion. Nevertheless, given that the Introduction section would become too short, we have kept it as a single section, but we have rewriten it for better clarification.
>
> > Q: The authors imitate two different sorts of noises: Poisson and Gaussian. While the previous works are mostly focused on Gaussian Noises, the authors didn’t provide the reasons for choosing Poisson Noise to denoise.
>
> A: We thank the reviewer for helping us to clarify this point. In this work, we simulate noise by a combination of Poisson and Gaussian probabilistic models to account for the quantic nature of X-rays and the presence of thermal fluctuations in the detector, respectively [1,2]. We have added the following sentence to the Introduction: *However, a reduction in dose implies an increase in image noise, which arises due to the quantic nature of X-rays and the presence of thermal fluctuations in the detector*. [...] *Poisson noise, which is the type of noise inherent to X-rays due to their quantic nature [...] is simulated using equation 4, which includes a small Gaussian noise $\eta ∼ N (0, I)$ scaled by $\sigma_2 = 10$ to emulate electronic noise.*
>
> > Q: The proposed method was only tested on one dataset: the NIH Chest X-ray database, with a random selection of 1225 images in total. There are two concerns regarding this: 1. The authors should try on different datasets. 2. Why not use the full dataset instead of sampling?
>
> A:
> 1. We agree with the reviewer on the importance of validating this method on a different dataset. Therefore, we have selected 22 noise-free pneumonia images from the [COVID19, Pneumonia and Normal Chest X-ray PA Dataset](https://data.mendeley.com/datasets/jctsfj2sfn/1), to show the robustenss of our method against pathologies not included in the training set. From the [qualitative results](https://imgbox.com/DCO8tDke) we can reach the same conclusions as for the NIH Chest X-ray dataset, i.e., that BM3D oversmooths the results, Nei2Nei does not achive complete denoising, and that DDPM-X preserves spatial resolution.
> We are working on the metrics values and eventually include them in future work.
>
> 2. One of the main goals of this work is to achieve denoising by training on a small database. Deep Learning models require large databases which are generally difficult to obtain in clinical scenarios. Additionally, pre-trained models available in the medical imaging field usually have a poor transferability and the efficacy of their reuse is limited [3]. To facilitate the implementation of these models in the clinics, demonstrating their utility when trained with small databases is of utmost importance.
>
> > Q: Are the training timestep and the testing timestep identical? e.g. 3 and 4 in low dose, and 8 in high dose? Is there any possibility to combine the two models regarding Gaussian and Poisson noises to perform a better result? So that in real use, we don’t have to choose from these two.
>
> A:
> 1. In the Evaluation section's second paragraph, we indicate the parameters used to simulate Poisson and Gaussian noise for the high and low doses in Table 2. We use the flood value *I* for Poisson noise, while for Gaussian noise, we use the noising timestep *t* from the Diffusion FDK. For the low dose case, Gaussian noise was simulated for t=9, while for the high dose it was simulated for t=3. Therefore, as discussed in the text, the denoising timestep t’ did not always coincide with the noising timestep t. We hope we make this clearer with the following modification of paragraph 2 of the Evaluation section:
> *Noisy images were obtained for a high dose ($I = 5 × 10^4$ and $\sqrt{(1 − α_3)} = 9.6 × 10^{-3}$ for Poisson and Gaussian noise respectively) which corresponded to an estimated denoising step of t′ = 3, and for a low dose ($I = 9 × 10^3$ and $\sqrt{(1 − α_9)}= 9.6 × 10^{-2}$), which corresponded to an estimated denoising step of $t^{'} = 9$.*
>
> Any comments are welcome to clarify this point further. If the reviewer considers it helpful for the reader, we may include this information in Table 2.
>
> 2. We apologize for the misunderstanding. A single model is used to denoise Gaussian and Poisson noise; therefore, no a priori knowledge is required to apply it, except the noise level. We have added and introductory paragraph in the Materials and Methods section to better explain our method and to hopefully clarify this point.
>
> ---
> [1] Qiaoqiao Ding et al. Statistical image reconstruction using mixed poisson-gaussian noise model for x-ray ct. 2018.
>
> [2] Xin Yi et al. Sharpness-aware low-dose ct denoising using conditional generative adversarial network. 2018
>
> [3] Yuncheng Yang et al. Pick the best pre-trained model: Towards transferability estimation for medical image segmentation. 2023

---

### Official Review · Reviewer_Z5PU · 2024-02-29

**Confidence:** 3
**Preliminary Rating:** 3
**Final Rating:** 4

**Summary:**

In this work, the authors propose a diffusion probabilistic generative model for Chest X-Ray image denoising. The authors show that their approach surpasses the performance of baselines.

**Strengths:**

1. A potentially interesting approach, noise reduction is an important problem for the X-Ray community as much as the use of diffusion probabilistic models.
2. The performance exceeds the baselines, and the authors show it by using 3 different metrics.

**Weaknesses:**

1. This paper can be benefited from better writing, especially in the Stage II of their approach.
2. This paper may need some elaboration on the more mathematical explanations.
3. Baselines can be expanded.

**Detailed Comments:**

1. I wasn't able to grasp the procedure for finding the Gaussian noise equivalent in Stage 2. Do you initiate the diffusion procedure not from $x_T$ but from some $x_{t-1}$? For the sake of completeness, unless there is a misunderstanding, could you formally describe how to find the equivalence between $Z_{log}$ and some $x_{t-1}$?
2.	What does the parameter $t$ correspond to in the second paragraph of Section 2.3? Does it describe where exactly you start the RDK?
3.	Can you also run a U-Net and vanilla Noise-2-Noise baseline? What was your motivation for running Nei2Nei model as the baseline?
4.	“1024x10204”
5.	Will you release the code upon acceptance?
6. Try to display the SSIM metric with 4 decimal places for convenience.

**Justification Of Final Rating:**

Adding the variance matching strategy to the DDPM-based planar x-ray image denoising model indeed enhances the value of this work. Furthermore, the authors have notably improved the writing and incorporated baseline comparisons and supplementary experiments to illustrate the strengths and weaknesses of their approach. While the experiments could potentially include a downstream task and discuss the runtime of the methodologies, in its current form, I still believe that this paper will be an important contribution to MIDL 2024. Therefore, I am raising my rating to 4 and recommend accepting this paper.

**Justification Of The Preliminary Rating:**

I have given a borderline rating since I wasn't able to understand some of the parts of the paper and the baselines appear to be expanded. I would be happily increase my rating depending on the result provided by the authors.

**Questions To Address In The Rebuttal:**

Mentioned in the detailed comments section.

---

> ### Author Response · Authors · 2024-03-15
>
> We thank the reviewer for their constructive feedback. We have rewritten the Materials and Methods section to better explain all stages of our method.
>
> > Q: I wasn't able to grasp the procedure for finding the Gaussian noise equivalent in Stage 2. Do you initiate the diffusion procedure not from $x_T$ but from some $x_{t-1}$? For the sake of completeness, unless there is a misunderstanding, could you formally describe how to find the equivalence between $ Z_{log}$ and some $x_{t-1}$?
>
> A: We agree with the reviewer that the description of Stage II may be convoluted, and we welcome the suggestions to make our explanations clearer. In this work, Gaussian noise is being simulated with the Forward Diffusion Kernel (FDK) for a specific timestep t<T. Therefore, we take that same timestep as the starting point for the denoising pipeline. In this way, we reduce the number of timesteps from T to t and highly speed up inference. We have rewritten the text as follows: *we follow a similar aproach to the Come Closer Diffuse Faster algorithm (CCDF). The RDK is applied from t = t′ to t = 0 (Fig. 1), where t′ is the denoising timestep obtained from an estimate of the noise level of the image. [...] Gaussian noise is simulated with equation 2 for a specific timestep t, and therefore t′ = t.*
> Furthermore, we have included a formal description of the pipeline followed to find the denoising timestep for Poisson noise, which is shown hereinafter.
>
> *Due to the signal dependency of Poisson noise, the denoising timestep $t'$ is estimated from the maximum noise variance found in the image, as follows*:
> \begin{equation}
>     \hat{Y_{log}}=\mathbf{I} * \frac{1}{n} \sum^N_{i=1} P_{99} (\max(Y^i_{log})) , \quad \mathbf{I} \in \mathbb{R}^{h\times w}
> \end{equation}
> *where $N$ correspond to the size of the training dataset. The percentile is applied to avoid the contribution of high intensity artificial details present in the images such as medical annotations. Then, $\hat{Z_{log}} $ is computed from $\hat{Y}=Ie^{-\hat{Y_{log}}}$ by using Equation 4, and the denoising timestep $t'$ is estimated as follows*:
> \begin{equation}
>     t' := 1- \overline{\alpha_t'} \thickapprox Var (\hat{Z}_{log})
> \end{equation}
>
>
> > Q: What does the parameter t correspond to in the second paragraph of Section 2.3? Does it describe where exactly you start the RDK?
>
> A: We apologize for the typo. As the reviewer has correctly identified, parameter t should be t’ and it corresponds to the denoising timestep, or equivalently, to the starting point in the RDK.
>
> > Q: Can you also run a U-Net and vanilla Noise-2-Noise baseline? What was your motivation for running the Nei2Nei model as the baseline?
>
> A: The reason for selecting Nei2Nei as one of the baselines was that, out of the supervised DL methods tested on Poisson noise, it is one of the most recent (2021) and a well-established supervised denoising method (more than 200 citations).
>
> We are working on adding a U-Net baseline and will show some preliminary results by the end of the discussion period. We tried to evaluate Noise2Noise but found that the official repository is not reproducible. To compensate for this, we show the results for DU-GAN [1], a more recent baseline requested by another reviewer that likely shows better results than Noise2Noise or a vanilla U-Net. Here we only show the validation for low dose, but we will include the high dose case by the end of the discussion period.
>
> From a [qualitative point of view](https://imgbox.com/CGjvOYj8), DU-GAN achieves better denoising than Nei2Nei, and although it does not smooth the image as much as BM3D, it still introduces blurring, specially in low contrast borders. This is likely the cause for the high value of LPIPS shown in the table below. Importantly, DU-GAN introduces halluciantions, as indicated by the red arrows.
>
> | Model   | t' | LPIPS    | SSIM      | PSNR      |
> |---------|----|----------|-----------|-----------|
> | BM3D    | -  | 0.05     | 97.81     | 34.58     |
> | Nei2Nei | -  | 0.03     | 96.85     | 36.45     |
> | DU-GAN  | -  | 0.04     | 97.61     | 37.04     |
> | DDPM-Gaussian | 8  | **0.02** | 97.33     | **37.27** |
> | DDPM-Poisson | 8  | 0.02     | **98.05** | 36.89     |
>
> **DU-GAN baseline will be included in the manuscript once the validation for the high dose case is completed.**
>
> > Q: Code release.
>
> A: Currently, we are working on a complete version of the code that includes multiple functionalities before releasing the code.
>
> > Q: Display the SSIM metric with 4 decimal places for convenience.
>
> A: We decided to use 2 decimal places to facilitate the tables' readability and avoid clutter. If 4 decimal places are to be used, we believe they should be added for all metrics, not only the SSIM. Nevertheless, if the reviewer deems it appropriate, we will implement this suggestion upon reply.
>
>
> [1] Zhizhong Huang et al. Du-gan: Generative adversarial networks with dual-domain u-net-based discriminators for low-dose ct denoising. 2022

---

> > ### Author Response · Authors · 2024-03-27
> > **Update**
> >
> > **DU-GAN and vanilla UNet have been included in the manuscript**. The UNet selected was the same as the one used by the proposed method, trained in a supervised fashion with the MSE loss. We have included the quantitative results in Table 2 and the qualitative results in Figure 2.

---

### Author Response · Authors · 2024-03-20
**Thank you for your feedback**

Dear reviewers,

We believe we have adressed all the comments of the reviewers. We have rewritten the Materials and Methods section to clarify Stage I and Stage II of our method, and have included an introductory paragraph to facilitate the comprehension of that section. We have also expanded the Discussion section to include the issues raised by the reviewers. Aditionally, in our rebuttal we have presented results regarding extra baselines and evaluation databases which will be included in the manuscript as soon as the experiments are completed.

In light of these changes, we await for your feedback.

---

### Author Response · Authors · 2024-03-26

Greetings

We have still not received an answer to our rebuttal. We believe we have addressed all points, and we would like to know whether if we will have enough time to address further comments.

Thank you
Regards

---

### Meta-Review · Area_Chair_zfEW · 2024-04-02

**Recommendation:** Accept (Poster)
**Confidence:** 4

**Metareview:**

After revision, there is reviewer consensus that the work can be accepted. Despite this positive outcome, overall enthusiasm was moderate largely due to small dataset sizes and strong focus on synthetic experiments. The real data experiments on an anthropomorphic phantom are considerably less convincing and small scale, making this paper more borderline that it seems to be from just considering the review scores.

---

### Decision · Program_Chairs · 2024-04-06

Accept (Poster)